# Closing the parachute and opening the umbrella: Strategies for inclusivity and representation in producing impactful coastal ecosystem research

## Perspective

community science; international collaboration; traditional knowledge; parachute science; colonial science

**Corresponding author:**
Katie May Laumann;
Email: klaumann@umces.edu

Katie May Laumann[1], Nicholas M. Hoad[2], Lauren Alvaro[3], Shahrzad Lili Badri[1], Noirin Burke[4], Annie Carew[1], Guilherme N. Corte[5], Aldo Croquer[6], Yasmina Shah Esmaeili[5], Martha Farrell[7], Naoko Kouchi[8], Juhyung Lee[9], Masahiro Nakaoka[10], Lina Mtwana Nordlund[2], Rita I. Sellares-Blasco[11], Ed Sheldon[12], Maria F. Villalpando[11] and Jonathan S. Lefcheck[1]

[1]Integration and Application Network, University of Maryland Center for Environmental Science, Cambridge, MD, 21613, USA; [2]Natural Resources and Sustainable Development, Department of Earth Sciences, Uppsala University, SE-621 57, Visby, Sweden; [3]National Sea Grant College Program, Silver Spring, Maryland; [4]Galway Atlantaquaria, Galway, Ireland; [5]Department of Marine Biology, College of Marine Sciences and Maritime Studies, Texas A&M University, Galveston, Texas, USA; [6]The Nature Conservancy, Caribbean Division, Punta Cana, La Altagracia, Dominican Republic; [7]Maharees Conservation Association CLG, Co Kerry, Ireland; [8]Amamo Works, Akkeshi, Hokkaido 088-1114, Japan; [9]Department of Oceanography and Marine Research Institute, Pusan National University, Busan 46241, South Korea; [10]Akkeshi Marine Station, Field Science Center for Northern Biosphere, Hokkaido University, Hokkaido 088-1113, Japan; [11]Fundación Dominicana de Estudios Marinos, Bayahibe, Dominican Republic and [12]Tralee Bay, County Kerry, Ireland

## Abstract

Parachute science is the problematic and extractive practice of non-local researchers taking data, knowledge and information from communities of which they are not members, failing to engage the local community and local scientists, marginalizing them in most aspects of the research, and using the results to their own benefit. Perpetuated by colonialism and unequal access to resources such as funding, education and data, it is harmful to local scientists and undervalues the contributions of the community as a whole. Ultimately, it erodes trust within the scientific community and, more broadly, builds dependence on foreign researchers and makes science less global and collaborative. Increasing international and cross-cultural collaborations while being careful to avoid parachute science can help minimize these impacts. Here, we offer our perspectives on parachute science and suggestions on how to avoid it, based on our experiences conducting research internationally with diverse scientists and communities, including both academics and non-academics. Instead of a parachute, we suggest opening the scientific "umbrella" to incorporate diverse perspectives and local contributions in generating relevant and impactful scientific insight.

Resumen.

La ciencia de paracaídas es la práctica problemática y extractiva de investigadores no locales que toman datos, conocimientos e información de comunidades de las que no son miembros, no logran involucrar a la comunidad local y a los científicos locales, los marginan en la mayoría de los aspectos de la investigación y utilizan los resultados para su propio beneficio. Perpetuado por el colonialismo y el acceso desigual a recursos como la financiación, la educación y los datos, es perjudicial para los científicos locales y subestima las contribuciones de la comunidad en su conjunto. En última instancia, erosiona la confianza dentro de la comunidad científica y, en términos más generales, genera dependencia de los investigadores extranjeros y hace que la ciencia sea menos global y colaborativa. Aumentar las colaboraciones internacionales e inter-culturales, al tiempo que se tiene cuidado de evitar la ciencia de paracaídas, puede ayudar a minimizar estos impactos. Aquí, ofrecemos nuestras perspectivas sobre la ciencia de paracaídas y sugerencias sobre cómo evitarlo, basándonos en nuestras experiencias realizando investiga-ciones a nivel internacional con diversos científicos y comunidades, incluidos académicos y no académicos. En lugar de un paracaídas, sugerimos abrir el "paraguas" científico para incorporar diversas perspectivas y contribuciones locales en la generación de conocimientos científicos relevantes e impactantes.

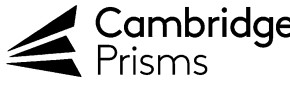

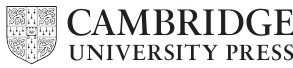

## Impact statement

Parachute science is the act of conducting research without acknowledging or engaging local communities. This practice is exploitative and diminishes the impact and value of science for

understanding, protecting and restoring coastal ecosystems. While parachute science has deep roots in colonialism, it can be perpetrated by anyone who fails to incorporate local context and perspectives in their science, even in their own country. In this article, we assembled a diverse international group from across the Global North, including academic, non-profit and volunteer scientists, to briefly review the topic of parachute science and to share experience-based strategies to avoid parachute science and eliminate this practice in one's own science. We provide strategies that will enable the researcher, instead, to engage and work with local communities globally. We aim to share the joy and added value we have experienced from, along with the strategies to enable collaboration with local researchers and communities in research. We hope that readers will appreciate the value gained by opening their science to local input and contributions under an inclusive umbrella. Furthermore, we would welcome a similar paper with actors in the Global South to provide their view on parachute science.

## Introduction

Imagine an adventurous and curious scientist. They jump from a plane and parachute to the ground with equipment to survey coastal habitats, their knowledge from years of academic training and funding to assess coastal resilience. They land on a litter-free beach; to one side are dunes dotted with tufts of grass and a row of houses just beyond. On their other side is a bay teeming with life: rock pools filled with seaweeds, a large seagrass meadow and many invertebrate and fish species. The scientist conducts an ecosystem survey and determines that the system is in excellent condition, given the rich biodiversity, lack of visible human disturbance and extensive habitat available across land and sea. They conclude that the system provides strong protection to the local human community from the threats of climate change. Having gathered data, the scientist proceeds to the airport to fly home. From their home office, they analyze their data and write up their findings without sharing their results, data or paper with anyone local to their field site. The article they publish is highly regarded but does not acknowledge or include contributions by local scientists and community members. Throughout the process, the scientist neglected to involve local perspectives and knowledge – including from their peers in the local scientific community – who could have contributed to all stages of the work, from planning through publication to management and advocacy.

This is a simplified example of parachute science. Although the scientist collected useful data, published a well-cited article and furthered their career, they neglected something extremely important: the context and impact that could have been added by engaging the local community – including Indigenous peoples, residents, policymakers, managers and scientists – who have a vested interest in the ecosystem. They did not consider how the data they collected might reflect on the community, benefit from local engagement and generate impact beyond a publication in the scientific literature. The people who have intimate knowledge of the system and who stand to benefit the most from the research were excluded. The scientist behaved in a manner that was extractive, exploitative and potentially harmful to the community, and in doing so, missed nuances that would have helped them more accurately evaluate the system.

Because our parachute scientist did not engage those with local knowledge, they did not learn the beach was litter free due to a volunteer beach clean-up effort that morning, and erroneously assumed a lack of human disturbance. They missed out on learning that the seagrass meadow was previously much more expansive and historically supported many more species, including an endangered shark. Seaweed biodiversity appeared high but had recently been reduced due to an invasive species that outcompeted native seaweeds. Storm surge had increased due to the shrinking seagrass meadow and had eroded much of the beach. The scientist failed to learn that a group of dedicated volunteers was working to map and monitor changes to the seagrass meadow, members from local businesses gathered each weekend to plant dune grasses, and a small nearby university had spent over two decades studying these ecosystems. This context would have enabled better scientific insight, and the local community would have benefited from participating in, and linking their efforts to, the global research community.

This simplified of parachute science is based on events that occurred at the beginning of an international collaboration. Two co-authors traveled to visit field research sites that could potentially contribute to their study of coastal resilience. They worked with a local, "non-academic" scientist, who made arrangements for meetings and workshops with local conservation groups and the community as a whole. Upon visiting the potential research site, the two traveling researchers marveled at the natural beauty of the area, the lack of human disturbance on the beach, the biodiversity of the marine life in the tidepools and the extent of the seagrass bed present. From their initial observations, they might well have made the same assumptions as the parachute scientist in our example. However, they were able to gain a more accurate perspective through discussions with their host, multiple field days with a local conservation group and extended meetings with local community members, and were therefore able to avoid the mistakes of our parachuter.

The authors of this paper are a diverse international group from across the Global North, including academic (at multiple career stages), non-profit and community groups of volunteer scientists. Some of us have experienced being disregarded or sidelined by international and/or purely "academic" researchers. The academic scientists in this group have worked extensively with diverse communities internationally and have faced the challenges of conducting science without engaging in "parachute science." Additionally, some have been victims of the phenomenon on behalf of other international scientists. We offer this firsthand perspective on the topic of parachute science and best practices on how to avoid it, based on our own experiences coupled with best practices culled from recent literature.

## What is parachute science?

Parachute science has been defined as:

- "when foreign researchers swoop in, disregard people with on-ground experience and give little to no credit to local collaborators…." (Watson, 2021).
- "when scientists and/or NGOs from the Global North venture to the Global South to conduct research or deploy programs and fail to invest in, fully partner with, or recognize local governance, capacity, expertise, and social structures" (deVos and Schwartz, 2022).

- "the practice whereby international scientists, typically from higher-income countries, conduct field studies in another country, typically of lower income, and then complete the research in their home country without any further effective communication and engagement with others from that nation." (Stefanoudis et al., 2021).
- "where external researchers collect and analyze data on a challenge in or adjacent to a local community, yet little to no benefit is received by the community from the work." (Wowk et al., 2023).

While many of these definitions place emphasis on access to resources or geography (the "Global South"), we believe parachute science can be conducted anywhere, and by anyone, if they fail to engage, and appreciate and acknowledge the knowledge and contributions of locals. This failure is compounded when insight gained by the scientist is not returned to the local community. Here, we define "community" as the group of individuals who primarily live or work in the area where research is occurring, and who have a vested interest in its environment and ecosystems. We use the term "community scientists" to refer to people who do not possess advanced degrees in science but actively collect data, monitor ecosystems, carry out science-based restoration activities (often under supervision of trained academics), have local, traditional or Indigenous knowledge, and otherwise participate in the scientific process.

At its core, parachute science is extractive. We define parachute science as researchers taking data, knowledge, and information from communities of which they are not members without adequately engaging the local community and local scientists, and using it to their own benefit – for example, in publications that elevate their standing in academia. Parachute science is sometimes called "colonial science" because of its parallels to colonialism. Regardless of motivation, parachute science perpetuates damage to the communities from which it takes and diminishes the potential impact of the work being conducted (Haelewaters et al., 2021; deVos et al., 2023). Parachute science can occur anywhere, not just in countries that have been colonized, and is not only conducted by scientists from previously imperial nations. Parachute science is not unique to the natural sciences, but we have framed our example above to reflect the experiences of coastal scientists and communities. We believe that the best practices described here can be applied to any field of research.

## How does parachute science harm communities?

The motivation of the parachuter does not factor into whether damage is done (Genda et al. 2022. A scientist may conduct research in a community that is not their own out of an intention to help the scientific field, world, or community in which they are working. Intention, however, does not negate the damage that excluding local researchers and communities from engagement or partnership does. Neglecting to involve local scientists – community scientists and academic researchers – sidelines them from participating in the scientific process (Watson, 2021; Odeny and Bosurgi, 2022). This outcome may increase dependence on outside researchers, excludes locals from scientific discourse and recognition, and reinforces power imbalances – for example, by publishing findings in "pay-walled" journals that are inaccessible to the community (Stefanoudis et al., 2021; Watson, 2021; deVos and Schwartz, 2022). It may also result in competing interests or undermining of locally led and community-driven research on the same

issues (deVos et al., 2023; Rayadin and Buřivalová, 2022). Parachute science puts the agenda of the parachuter rather than the needs of the community foremost. Such practice may lead to significant harm to the ecosystem being studied, breaking of local regulations of which the parachuter is unaware, and underinformed conclusions that may lead to misguided action by others (Watson, 2021; de Oliveira et al., 2024).

In addition to contributing to the broad systemic problems mentioned above, failure to engage locals may result in incomplete or inaccurate research: incorporating local knowledge is often critical to a broad understanding of the location, system, or process being studied. Although community science contributions to research have increased in recent years (particularly in monitoring coastal habitats), it has been widely recognized that these contributions, and those of local academics, are underutilized and undervalued by traditional researchers conducting work in locales that are not their own (Champion et al., 2018; Baustian et al., 2019; Lucrezi 2021). Engaging community and/or local academic scientists can help fill data and knowledge gaps (Champion et al., 2018; Jones et al., 2018), particularly in countries where traditional academic research is not as active or well-funded; provide more consistent and comprehensive on-the-ground context for rapid environmental changes (Reyes-García et al., 2020), and strengthen research and project design to best fit the local ecological and social context (Alexander et al., 2019).

Avoiding parachute science must go beyond just paying local people for data collection; it must meaningfully engage them in the research. This ideally means including them throughout the research process, from conceptualization, planning and implementation, to analysis, interpretation, and publication of research findings. A scientist visiting for a month simply cannot acquire the same perspective as those who are living in and among the resource. When engaging with locals, researchers must remain vigilant to avoid extractive practices. Indigenous communities, for example, have collected and managed data since long before colonization and these communities have rights over how these data are used, stored and shared (Hudson et al., 2023). Without strategies in place to form ethical partnerships based on collaboration, and without making this sovereignty explicit within data, these rights are easily disregarded. Furthermore, such data can end up in large, amalgamated repositories without attribution and context, compounding many of the other issues mentioned already (Jennings et al., 2023).

## Drivers of parachute science

Parachute science largely results from historical power imbalances and its continued practice, facilitated by internal complicity, perpetuates these imbalances. Drivers of parachute science include: "wealth and opportunity disparity" (deVos and Schwartz, 2022; Meyer-Gutbrod et al. 2023), language/cultural differences stemming from English being the default language of scientific discourse (Rayadin and Buřivalová, 2022, deVos and Schwartz, 2022), access to resources (deVos and Schwartz, 2022), and difficulty building trust (Watson, 2021; Odeny and Bosurgi, 2022). Other drivers may include underutilization of non-academically produced science (Cigliano et al., 2015), bias against perceived "substandard" data collected by volunteers, inadequate effort to engage locals as equal partners, and reluctance to venture outside of the ivory tower. Power imbalances are compounded when funding is offered to foreign researchers who do not engage with communities.

Parachute science may also stem from ignorance: many scientists may be unaware that they are engaging in it, or of how to avoid it. In an informal survey given to 32 international scientists (including academic and non-academic scientists) at the beginning of an "Avoiding Parachute Science" workshop in 2024, 65% indicated that they were unsure or did not know what parachute science is (Laumann et al., 2024). Clearly, the practice of this exploitative form of science is not always obvious, even to the established researcher. Those who are aware of parachute science may commit it because they do not know how to avoid it, while researchers from diaspora communities, for example, may be well-placed to act as conduits between research institutions and communities, but may lack institutional support for this, as has, for example, occurred in the African diaspora (Mwampamba et al. 2022). The fact that parachute science is not always easily recognizable or easily avoidable highlights the importance of recognizing what parachute science is as an individual, and the value of developing support structures to reduce unethical practices (Mwampamba et al., 2022).

## Why and how to stop parachuting and open the umbrella instead

Engaging with community scientists can maximize the value of science by addressing the limitations of academic endeavors, particularly those related to time, effort and cost. Local and community scientists are an often-overlooked potential partner in overcoming these limitations (Cigliano et al. 2015, Genda et al. 2022). When properly engaged, they can provide essential support and insights. For instance, enlisting recreational divers (Branchini et al., 2014; Edgar et al., 2020) or anglers (Champion et al., 2018) enables scientists to gain a broader perspective on coastal ecosystems, ensuring the success of scientifically-driven conservation and restoration efforts (Bird et al., 2014). Volunteers have been an essential part of collecting seeds for one of the largest and most successful seagrass restorations worldwide (Orth et al., 2020). Volunteer divers and snorkelers have propagated and replanted threatened coral species in the Caribbean, with a success rate comparable to efforts by scientific experts (Hesley et al., 2017). The participation of community scientists in collecting data on mangrove "blue carbon" has allowed Belize to lead the way in addressing climate change (Morrisette et al., 2023). Providing community scientists with training can build capacity within the community and support continued or future science done by local academics. Aside from the monitoring benefits, actively engaging with interested locals increases their investment in environmental conservation and ensures that vitally important economic and social concerns are addressed in application of the science towards management/policy (Baustian et al., 2019; Dean et al., 2018; Vargas-Nguyen et al., 2020; Sellares-Blasco et al., 2022).

Many solutions to avoid practicing parachute science have been proposed by scientists and the individuals impacted by it. Top-down controls that safeguard against the practice of parachute science include: rejection of publications that do not include equal contributors as authors; requiring researchers to indicate that they have received permits to conduct the research being published and encouraging authors to discuss efforts taken to avoid parachute science (Stefanoudis et al., 2021; Watson, 2021). A Contributor Role Taxonomy or CRediT statement defining the roles of authors can help highlight unrecognized contributors who have made a substantial impact on the project and who should be formally recognized in the output. Funders may require that proposals include local community partners in research, and require compliance to ensure those researchers are benefiting from the work (Watson, 2021; deVos, 2022). Adoption of frameworks like The Convention on Biological Diversity's Access and Benefit-Sharing (ABS, Secretariat of the Convention on Biological Diversity, 2022) frameworks can be effective in curbing parachute science. ABS frameworks make it essential to identify suitable collaborators in each country, establish a memorandum of understanding between institutions, obtain research permits for foreign researchers, and fulfill additional requirements when conducting coastal research in Southeast Asia. Adhering to these procedures is a first step toward preventing parachute science.

While top-down controls address some aspects of parachute science, they are not universally practiced. Therefore, the burden of ensuring we scientists do not perpetuate this practice falls on us: it is up to us to close the parachute and open an umbrella to include and recognize a diversity of scientific contributions.

A list of best practices to avoid parachute science is below. It contains suggestions based on the experiences of the authors and from the literature: it is by no means exhaustive. Ultimately, parachute science is a behavior, and it is up to the individual scientist to avoid this behavior. Other authors (e.g., Stefanoudis et al., 2021) have provided best practices, some similar to our own, and we do not seek to replace their suggestions. Rather, we build on them. The unique inclusion of non-academic scientists among our authors provides an expanded view of best practices, including novel suggestions while building on and bolstering some that have been suggested by previous authors.

1. Perhaps the most obvious but essential piece of advice is to *identify and engage with local scientists and communities*. Ideally, all scientists know they should be doing this, but in practice engagement can be difficult, starting with simply identifying potential collaborators. Sometimes local partnerships emerge organically through mutual acquaintances or colleagues, but most require additional effort (Genda et al. 2022). Local scientists may lack relationships with non-academic communities in their research sites. When this occurs, it can be challenging for foreign researchers to effectively engage communities. Therefore, selecting the right local collaborators is essential. One strategy is to collaborate with not only natural scientists but also researchers actively engaging with local conservation and restoration efforts, leading community-based science projects, or studying social-ecological systems (Hemmerling et al. 2019; Nakaoka et al., 2018). Collaborating with researchers already integrated with the local community can broaden impact and enhance local capacity for locally led or international collaborative research (Sellares-Blasco et al., 2022).

2. *Get out of the "Ivory Tower" of academia.* Do not restrict your discourse of science to other academic scientists. The authors have found success in identifying and engaging with local communities and nonprofit collaborators by searching for and messaging local groups interested in their work and operating in the study area through their official websites or social media accounts is a potential path toward engagement. Contacting local, state, regional or national universities and/or management authorities has also yielded valuable contacts. Online nature-based community groups, such as iNaturalist, have already self-selected for participants who are passionate about the environment and provide a direct conduit to those already generating local verified observations.

Increasing communication with or participation of local communities in research ensures they are part of the narrative of research in their own community. It supports the co-design of research, providing collective benefits for the community and enhancing the quality and relevance of the science.

3. *Abandon the "Savior Complex."* Researchers may come across as patronizing when they offer "help." Instead of saying "I have come here to help you," ask "how can my work benefit you?" or "what can I do to contribute?" Empower without elevating yourself above others. Often this requires taking a back seat and supporting local efforts without dominating the narrative. Operating in a supportive role gives others the chance to have ownership of the research, which often results in more successful outcomes (Gillgren et al., 2019; Moore and Kumble, 2024).

4. *Engage early.* Prior to arriving in a community, and ideally prior to developing funding and research proposals, organize a community workshop to explain your research to locals, and more importantly to listen to their feedback, point of view, and concerns. Identify priorities or goals you share with the community and be prepared to revise your own goals to include theirs. Be open to their participation in your work, regardless of their training: expertise does not require a degree. When scientists listen before action (Singeo and Ferguson, 2022; Spencer et al. 2023), they open themselves to other ways of understanding ecological systems. Informed prior engagement allows for research to be designed for mutual benefit which can meet local needs, complement ongoing efforts and benefit the communities where research occurs.

5. *Recognize when and how different contributors can (and cannot) add value.* Many aspects of science, such as planning and data collection, are enhanced through participation of community scientists. In some cases, data analysis and interpretation are best left up to trained academics who are able to remain unbiased in their assessments. However, when local, academic scientists or research-based groups already have expertise in data analysis or have expressed desire to learn, it is important to integrate them into the scientific process and provide opportunities for training that will build capacity. While certain analyses may require equipment available only in developed countries, inviting graduate students from host countries to receive training and perform these analyses provides opportunities for knowledge transfer. Although this approach demands significantly more time and resources compared to analyzing data ourselves, it contributes to the advancement of science and capacity building in host countries. It fosters the development of the next generation of scientists, ensuring a sustainable and equitable future for international research.

6. *Establish common goals and expectations.* When engaging communities and potential collaborators, be clear about project objectives, how the community will benefit from collaboration, and the value that would be added by the collaborator(s). Listen to the needs of the collaborators and adjust your plans accordingly. Rarely do local partners value the same outputs as scientists and their universities, namely, grant dollars acquired and the number/impact factor of scientific articles. Instead, they may desire other end products, such as skill development and training, analytical support, technical reports, "report cards," and social media content. However, non-academic collaborators may benefit from being listed as co-authors, as both formal recognition for their contributions and a path toward building their status as trusted experts in their community.

7. *Sustain communication.* Engage with collaborators often, including after work is completed (Ruppert et al. 2022). Once on-the-ground research is complete, scientists should maintain communication with local groups, share the data with them, and allow for their input, feedback and edits to the publication (Chouinard et al. 2008; Sellares-Blasco et al., 2022). Acknowledge them in presentations regarding the work. Strive to help communities or organizations promote the research through less academic channels, for example, co-author an article for a local newspaper. Continued contact beyond the life of the research is a sign that parachute science is being effectively avoided, and may lead to lifelong friendships. Demonstrating a long-term commitment, rather than merely gathering information and leaving, shows genuine respect and ensures that communities have science-based information that they can use in promoting restoration or conservation actions (Sellares-Blasco et al., 2022). Although funding opportunities may not support this, the researcher should build it into their regular practice.

8. *Be creative about shared funding.* When possible, local scientists should be included in fundraising. Sub-awards or shared funding can go a long way in supporting their participation. When shared funding is not possible, honorariums can defray hardship incurred by participating in research. When financial resources cannot be shared, providing food, transport, and apparel with the institutional logo can be a way to show appreciation. If possible, the non-local scientist may provide basic equipment that will enable the community to continue the research effort themselves.

9. *Understand and accept local norms.* This may mean collaborators should be addressed by their title rather than their name. It may require scheduling of fieldwork accounts for differing needs of communities, such as prayer, fasting, or other cultural or religious practices or events. Local customs must be respected, and when invited, the non-local scientist may engage in them respectfully. Learn how to greet someone, say thank you and goodbye in the local language. When not speaking the local language, speak slowly and clearly and avoid using uncommon words that may be unfamiliar. If the collaboration occurs over extended periods, consider learning the local language. This shows the utmost respect and provides for a more genuine connection and trust with local communities, which can lead to greater collaborative success.

10. *Share data and expertise.* Make data available in a standard format in open repositories and directly share data with researchers and local communities. Data should remain accessible; file formats that require expensive software should be avoided. Collaborators may not have access to high-speed internet to acquire large files from the cloud; in this case, mailing a hard drive may be the most efficient way to share data. These actions can indirectly reduce the risk of parachute science by removing some of the barriers that contribute to opportunity disparity (Stefanoudis et al., 2021; deVos and Schwartz, 2022). Similarly, publishing articles as Open Access and writing a companion piece for a general audience in the local language, such as in the model of *The Conversation*, can help make science more broadly accessible. Companion pieces in newspapers, local (to your study site) newsletter,

and magazines, as well as providing interviews to local podcast and radio programs when possible, can expand the reach of research and its benefits to the community. While some journals and funders have instituted policies enacting these controls, the real solution to parachute science relies on parachuters themselves making an effort to change the way they conduct science.

11. *Acknowledge data sovereignty.* It is vital to obtain consent from the community regarding storage and use of data. Non-local scientists should be aware of the rights of the community regarding data governance, such as indigenous Data Governance protocols. Guiding principles for data governance are becoming formalized and can provide a framework to which all researchers can look to in order to avoid parachute science practices. Specifically, the Collective benefit, Authority to control, Responsibility, Ethics (CARE) principles ensuring ethical handling of indigenous data (Carroll et al., 2020, 2021; Jennings et al., 2023) should be used by all researchers and organizations seeking partnerships with indigenous communities. Such principles can guide work in other contexts to develop ethical practice and equitable partnerships.

## Conclusion

Most academics recognize that the time for practicing science without considering non-traditional knowledge, community needs, or the impact of research is over. Although parachute science is widely accepted to stem from colonialism and may be perceived as primarily being conducted by academics from developed countries leveraging their resources to conduct work in under-resourced countries, this practice can also be carried out by researchers from developing nations or even scientists acting within their own borders. The challenge in confronting parachute science is that not only can it be difficult, it is a personal choice to be made by individual scientists. It cannot be solved by journalistic policies alone, or by a few individuals; it must be broadly adopted across academia.

As science has become more global, some practitioners have moved away from parachute science, yet many are still unaware of the dangers that parachute science poses for both the field and for the people who rely on resources under investigation. The contributions of community scientists have long been overlooked by academia. There is no room for parachute science if science is to become truly, equitably global and diverse. If scientists are to be trusted to guide management decisions in our rapidly changing world with already limited capacity, we must quickly adjust our attitudes towards who can practice science and how we practice science as a global community.

While avoiding parachute science is difficult, the benefits are clear: it makes research relevant beyond academia (Sellares-Blasco et al., 2022); provides valuable context and knowledge; provides opportunities to build collaborations and friendships that can progress shared goals; and overall makes science a better and more enjoyable field in which to work (Stöfen-O'Brien et al. 2022). Avoiding parachute science cannot be passive. We must actively think about how we engage with local communities and scientists from the time we conceive of a research idea and beyond publication. We must commit to opening the umbrella of inclusivity and closing the parachute if our work is to be relevant and useful, and if it is to produce much-needed change in the face of unprecedented global environmental challenges.

**Open peer review.** To view the open peer review materials for this article, please visit http://doi.org/10.1017/cft.2025.10004.

**Data availability statement.** The thesis of this publication is based on the experiences of the authors and the literature, which is cited. Data from informal, anonymous surveys mentioned are available by contacting the corresponding author.

**Acknowledgements.** We acknowledge all the people who have directly or indirectly contributed to the ideas in this manuscript, including all of the participants in the two workshops held at the International Seagrass Biology Workshop in Naples, Italy, in June 2024, and at the NOAA Sea Grant Marine Debris Symposium Agenda in November 2024. We would also like to thank all researchers with whom we have collaborated worldwide, and T.C. Berlioz for lessons learned.

**Author contribution.** KML, SLB, NB, GNC, A. Croquer, NK, MN, LMN, RISB, ES, and JSL contributed to the conception of the paper. KML, LA, SLB, NB, A. Carew, A. Croquer, NK, JL, MN, LMN, RISB, ES MFV, and JSL contributed to the design and implementation of the work, including convening workshops and discussion groups critical for gathering perspectives presented. KML, LA, NB, A. Croquer, GNC, YSE, NMH, NK, JL, MN, LMN, RISB, ES, MFV, and JSL contributed to critical discussion of parachute science, development of best practices for avoiding it, and experiences in international and citizen-driven research which were directly reported in the paper.

KML, JSL, NMH, GNC, and A. Carew drafted the paper. All authors helped author the paper, including contributing sections and providing critical revisions.

**Financial support.** This work was supported by a New Faculty Research Grant of Pusan National University, 2023 (no grant number). NH and LMN were supported by the Swedish Research Council (grant number 2021-03773).

**Competing interests.** The authors declare no conflict of interest.

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
