## [Reviewer Report]

Review, « Parachute science » from Katie May Maumann et al. 2025

This short paper reviews the practice of the so called parachute science practice and propose a review of experience-based strategies to avoid it.

The intro begins with an illustrative utopian tale that leads to understand and define parachute science and its pitfalls.

The authors then reviews how such practice harms communities, then analyze the drivers and finish with a review of strategies to stop parachute science.

The paper is well-structured, and personally, I like the views and ideas that totally make sense to me. However, I see two large flaws that prevent publishing in the present state:

1) The tone

2) The substance.

Specifically:

L68-82, the utopia is very relevant to the text. However, if it is totally made up from imagination, there is no point. The authors need to rely on a true story.

L90-92, I would provide two examples: one for Corals (the example in the utopia); one for Social or anthropologist science. E.g. similarly to natural science practice, an anthropologist may very well study a population and then leave and publish. This would avoid the intuitive thought that only natural sciences are prone to parachute.

All over: Watson 2021 is cited to justify many assertions. This is a (very interesting) “News Feature” and therefore not a peer review paper with data to provide arguments. I suggest to have it mentioned as a background, but It can’t be used to justify assertions.

L.133-135: “Regardless of motivation, parachute science perpetuates damage to the communities from which it takes and diminishes the potential impact of the work being conducted. » I agree with the statement, but it is not justified ? What project did that ? Publication(s) that identified such practices? How was that harmful to communities?

L145-146: reference for this statement?

L164-165: lessons from what?

L184: there is a reference to a workshop in 2024: please refer to report or say more in an appendix: what is this workshop, who was there, …

L187: diaspora communities: example?

L192-208: this paragraph is quite exemplary to me, with assertions substantiated by examples and reference – I am suggestion all other paragraphs have such structure.

L245-254: lacking references

L.232-345: the authors provide 10 good practices. They make sense (to me) and I think would be very useful for practices. However Stefanoudis 2021 does the same (see extract below)- the authors need to explain why they provide new good practices then.

Overall, I find the tone over-assertive, and given the lack of justification of the assertions, it may be perceived as messianic. The positioning of the authors with respect to their reader (the rest of the science community) gives the impression of messages coming from an “Ivory tower” group of authors who know better, delivering their opinion rather than facts (see best practice no2).

I suggest the authors review their assertions in their tone, eg “we must do” into “our review reveals that such practice is essential to attain …” and modify the use of references in a more specific way: (de Vos or Stefanoudis provide such arguments based on 10 if not 100’s of case studies); refer to cited paper conclusions (not cited authors alone, eg “based on xx studies, de Vos conclude that…).

In conclusion, what is written totally make sense to me, but it needs to be revised so that assertions are substantiated by facts. I recommend a major review.

Extract from Stefanoudis et al 2022):

Below are some recommendations to help stop parachute science. Similar sets of recommendations have been provided for the field of paleogenomics6, but here we focus specifically on the field of marine research. These recommendations are addressed to scientists conducting research overseas and research publishers. However, other sectors must also change their practices, including academic and research institutions, ethics committees, and funding bodies.

1 Find academic collaborators: start with host-country institutions with a national reach or scope. Online databases (Scopus, Web of Science) can help locate individuals and their work. Articles published in host-country journals (including university in-house journals) provide insights on potential collaborators’ expertise. An in-country visit and/or webinar early on in a project would help identify the most appropriate collaborators and are thus recommended.

2 Liaise with government funding bodies of the host nation: these can connect suitable collaborators, especially those with a track record for delivering on research grants.

3 Develop a joint research agenda: once appropriate collaborators have been identified and before fieldwork takes place, an extensive consultation with host-nation stakeholders is necessary so the research agenda is jointly framed and addresses local research needs.

4 Engage with the next generation of researchers: we strongly encourage the establishment of internship and exchange programs between partnering institutions for promising early-career researchers and/or co-supervision of students. This will not only provide enriching experiences for all parties involved but also, and most importantly, will help build and develop local talent and leadership that in time will be less reliant on foreign expertise.

5 Share academic literature: scientists from a high-income country working with colleagues in lower-income nations are encouraged to share copies of key papers from their personal collections, and where possible, make such personal collections available to local universities.

6 Know the regulatory landscape: many countries are very wary of specific research themes (for example, bioprospecting7). Regulatory bodies and agencies therefore have guidelines to vet applicants and applications. Partnerships are key in order to navigate requirements and provide useable information. Finally, many institutions also require better host-country engagement as part of research ethics approvals.

7 Transparency in publishing: journals should make it mandatory for authors to provide research permit and research ethics permit number(s). Editors and reviewers should confirm the existence of these or agree on a justification as to why one was not needed, in the same way that studies conducting experiments using animals require ethical approval, and which is clearly communicated in published articles.

---

## [Reviewer Report]

This manuscript addresses an important topic and the relationship between science, scientists and society. I have the following comments:

1. The paper opens with a hypothetical example of parachute science and its outcomes. I was disappointed to not see real-world examples and suggest the authors consider at least a table that documents examples in practice (good and bad) – the situation, the outcome, lost opportunity – that need not include specific locations/references in order to avoid any sensitivities. It would fit to complement the text at Line 151. This could also include examples where practices have led to outcomes that have led to demonstratable disadvantages to communities and/or environmental harm.

2. The first paragraph of the section “How does parachute science harm communities?” better belongs as a concluding paragraph to the previous section.

3. Line 163 – this should include participation in the full cycle of research – from planning through implementation to analysis and interpretation.

4. The authors address what individual scientists can do to alter their behaviour (line 229 onwards) and this is clearly stated as the principal focus of the manuscript. However, there is no discourse regarding the institutional (primarily funding mechanisms) setting that act as barriers to promoting behaviours to avoid parachute science, and how this should be changed (for instance eligibility criteria and other financial rules often effectively lead to parachute practices). I think it is important to address this. For instance, at Line 209 there are some examples of institutional solutions, but no mention of how institutional practice can perpetuate and effectively encourage poor practice.

5. Line 325, solution 10 – this could also include non-science sources, for instance, newspapers and other media outlets.

6. The manuscript is well written. However, there is some repetition between sections and the authors should check the text to remove this.

---

## [Editor Report]

This is a well written manuscript that covers an extremely relevant and important topic. I highly recommend that the authors, as both reviewers suggest, include real-world examples within and throughout the text where possible to ground the paper more fully and move away from hypothetical scenarios. This includes ensuring stronger justification of the arguments presented especially around demonstrable damage to communities as well as the diminishing of impact of the projects themselves. In addition, as both reviewers suggest, addressing the underlying funding implications and regulatory landscape within which parachute science practice is enabled or curbed will improve the manuscript. This will add more clarity to the purpose of the manuscript in providing a greater sense of understanding around determining and promoting behaviours that aim to avoid parachute science.

---

## [Reviewer Report]

I have reviewed the responses to my comments along with the responses to the other reviewer. I find that all the responses are appropriate and most often fully address the reviewer comments - and in the couple of instances where this is not the case the reasons provided are reasonable and should be accepted. This is primarily as a consequence of the article reflecting a perspective such that opinions upheld by the authors without necessarily having direct supporting evidence from other literature is reasonable, as well as a jsutified need to respect the author group and those invovled but not in the author group.

---

## [Editor Report]

The authors are commended for addressing the reviewer comments where appropriate and challenging the comments when necessary from the author group perspective. The changes to the manuscript have strengthened the text whilst remaining true to the author group experience and discourse.